# Asymbiotic germination and morphological studies of seeds of Atlantic Rainforest micro-orchids (Pleurothallidinae)

**Franciele Marx Koene, Érika Amano, Eric de Camargo Smidt, Luciana Lopes Fortes Ribas**  *

Departamento de Botânica, Setor de Ciências Biológicas, Universidade Federal do Paraná, Curitiba, PR, Brazil

* llfribas@ufpr.br

**Data Availability Statement:** All relevant data are within the manuscript.

**Funding:** The author(s) received no specific funding for this work.

## Abstract

The morphological and morphometric characters of seeds belonging to 11 species of the subtribe Pleurothallidinae using light and scanning electron microscopy were studied to understand the *in vitro* germination process. Qualitative data (color, shape, ornamentation) and quantitative ones were also evaluated in seeds and embryos (length, width, volume and air space percentage between the integument and the embryo). The viability of the seeds was evaluated by *in vitro* germination in woody plant medium (WPM), and by analysis of the developmental stages of protocorms until seedling formation (two to 24 weeks). Morpho-metric data showed variations within the genus *Acianthera* and between species of different genera. The best germination and protocorm formation responses occurred with *Acianthera prolifera* (92%) and *Acianthera ochreata* (86%), with the formation of seedlings after 12 and 16 weeks of sowing, respectively. The seeds and embryos of *A. prolifera* and *A. ochreata* were larger (length, width, and volume) with a structural polarity that may have facilitated their germination comparing to others studied species. Other characteristics of *A. prolifera* seeds that may have contributed to these results include the presence of a thin testa without ornamentation and a suspensor. The protocorms of *Anathalis obovata*, *Dryadella liliputiana*, and *Octomeria gracillis* developed slowly in the WPM, not reaching the seedling stage in 24 weeks of cultivation. This morphological and morphometric study contributes to the under-standing of asymbiotic germination of some micro-orchid species.

## Introduction

Orchids are suffering from an uncertain future through overexploitation, habitat loss due to human activities and the impact of climate change, and their survival is contingent on a variety of abiotic and biotic factors and their effect on orchid growth, development, and reproduction [1]. Their unusual physiology, seed structure, and germination pattern set them apart from other flowering plants. The seeds are dust-like, non-endospermic, and require a fungal stimulus for germination in nature [2, 3]. A unique characteristic of orchid seeds is that, rather than an endosperm, the air space surrounds a tiny globular embryo containing a small number of

**Competing interests:** No authors have competing interests

cells, which is protected by a membranous testa; the air-space volume in orchid seeds varies depending on the species [4, 5]. Seed morphological traits are thus related to biological and ecological processes like dormancy, germination, and seed dispersal [4].

Orchid propagation in nature is highly complex, involving specific mycorrhizal associations. *In vitro* methods, or plant tissue culture, can provide alternative approaches for both propagation and preservation, with asymbiotic germination representing an ideal system for studying the growth and development of seeds and seedlings [6, 7]. *In vitro* germination of orchids can help increase the effectiveness of conservation and breeding programs due to their high germination rates, which are commonly over 70% for epiphytic orchids, as opposed to under 5% in *ex vitro* conditions [8]. The culture media used for asymbiotic germination vary according to the species, and the most commonly used for the propagation of orchids are MS [9], VW [10] and KC [11, 12]. Koene et al. [13] recommended Woody Plant Medium (WPM) culture medium [14] for asymbiotic seed germination and plantlet development of *A. prolifera* when compared to MS, MS with half the salt concentration (MS/2) and KC. This culture medium was also better for germination and seedling development of other species of orchids, such as: *Brasiliorchis picta* (Hook.) R. B. Singer et al. [15] and *Hadrolaelia grandis* (Lindl.) Chiron & V. P. Castro [16].

The Orchidaceae is one of the largest and most diverse families of flowering plants in the world, with 25,000–28,000 species [2]. One of the subtribes of this family, Pleurothallidinae Lindl., belongs to Epidendroideae subfamily and Epidendreae tribe, consisting of approximately 5,100 species popularly known as micro-orchids [17]. This subtribe is found in the Neotropics, from Argentina to southern Mexico. Micro-orchids in the Atlantic Forest in Brazil, which is home to a large number of micro-orchid endemic species and is one of the most threatened extinction biomes on the planet, face extreme extinction pressures with a reduction in the original habitat of over 90% [18–20]. Most species of Pleurothallidinae have no great commercial appeal, due to the small dimensions of the plants and their flowers, as well as the difficulties of cultivation [18]. There are few studies of asymbiotic germination of micro-orchids, which have shown low germination rates and slow growth of seedlings [21, 22]. The best results were achieved by Koene et al. [13] with 79% of total germination of *A. prolifera* grown in WPM medium for 12 weeks. Studies on the morphology and morphometry of the seeds and their relationship with micro-orchid germination are also scarce.

Although the seeds of different orchid species have many similarities, there is significant variability in the size, shape, characteristics of the testa cells, the zones of adhesion, and sculptures constituted by the cellular wall and cuticular material [23]. Studies on seed morphology and morphometry have achieved substantial contributions to the taxonomy, phylogeny, and phytogeography of this group [24, 25]. Meanwhile, these studies can also enhance understanding of the *in vitro* germination process, which in turn can accelerate the production of seedlings for future reintroduction into their natural habitat, aiding in the conservation of the Pleurothallidinae. This study was carried out with 11 native species belonging to six genera of the Atlantic Forest (Fig 1) that have not been evaluated for the threat of extinction by the *Flora of Brazil* [26, 27], as follows: *Acianthera aphthosa* (Lindl.) Pridgeon & M. W. Chase, *Acianthera hatschbachii* (Schltr.) Chiron & van den Berg, *Acianthera ochreata* (Lindl.) Pridgeon & M.W. Chase, *Acianthera prolifera* (Herb. ex Lindl.) Pridgeon & M. W. Chase, *Acianthera sonderiana* (Rchb.f.) Pridgeon & M.W. Chase, *Anathallis obovata (*Lindl.) Pridgeon & M.W. Chase, *Dryadella lilliputiana* (Coan.) Luer, *Dryadella zebrina* (Porsch) Luer, *Octomeria gracilis* Lodd. ex Lindl., *Pabstiella fusca* (Lindl.) Chiron & Xim. Bols, and *Stelis grandiflora* Lindl. *Acianthera aphthos*a was the only species considered critically endangered (CR) in the Brazilian state of Espírito Santo by the Red List [28]. Five of these species are endemic to Brazil (*A. ochreata, A. aphthosa, A. sonderiana, D. liliputiana,* and *O. gracilis*) [26, 27]. *Acianthera prolifera* is

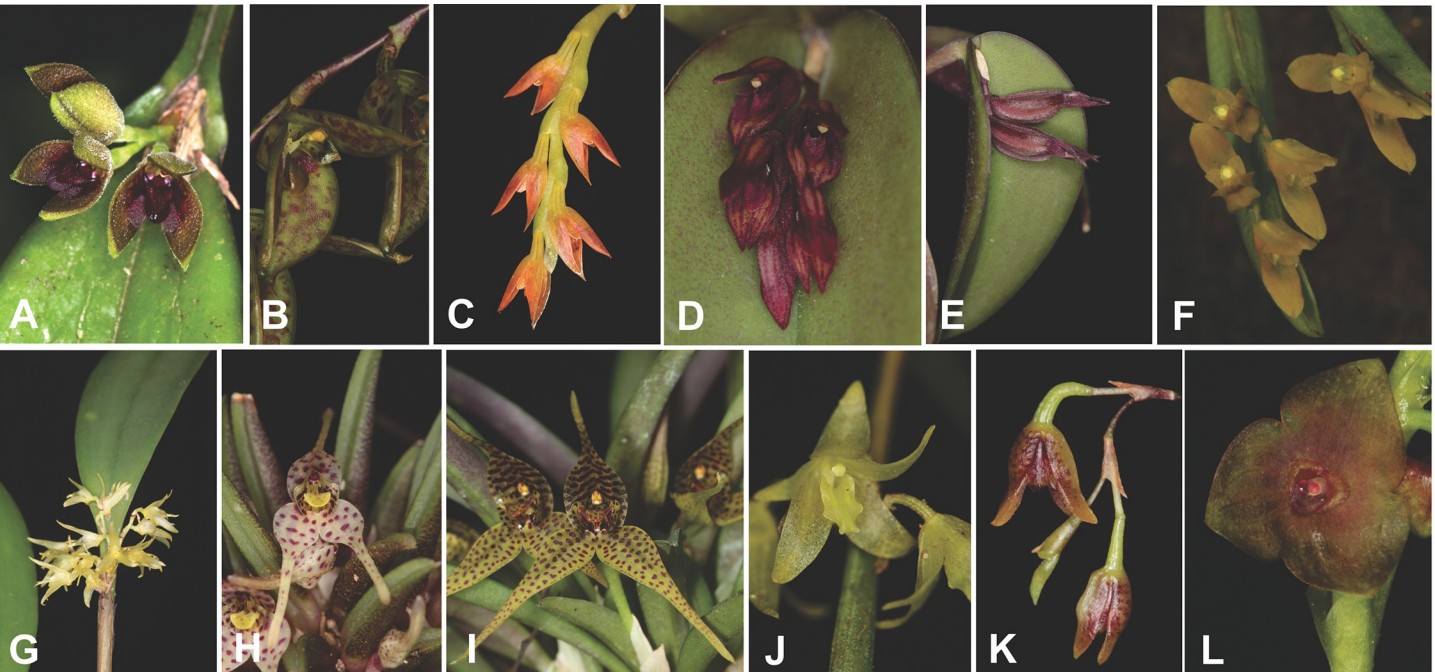

**Fig 1. Plants with reproductive structure of the studied species of the subtribe Pleurothallidinae. A.** *Acianthera aphtosa*, **B**. *Acianthera hatschbachii*, **C**. *Acianthera ochreata*, **D-E**. *Acianthera prolifera*, **F**. *Acianthera sonderiana*, **G**. *Anathallis obovata*, **H**. *Dryadella liliputiana*, **I**. *Dryadella zebrina*, **J**. *Octomeria gracilis*, **K**. *Pabstiella fusca*, **L**. *Stelis grandiflora*. Photos by Eric C. Smidt, except **F** by Luiz F. K. Varella.

rupicolous and *A. ochreata* is rupicolous and epiphytic, while all the other species in this study are epiphytes.

This study aimed to analyze the morphological and morphometric characteristics of the seeds and relate them with the asymbiotic germination of species of the subtribe Pleurothallidinae. We demonstrate efficient methods of rapid seed germination and seedling development of some species and our results offer potential possibilities for reintroduction programs that can play a key role in reducing the extinction risk for species of micro-orchids.

## Materials and methods

### Seed materials

Mature capsules at dehiscence from manual cross pollination of 11 species of the Pleurothallidinae subtribe (Fig 2A) were collected (Fig 2B) from three to four plants of each species in a greenhouse of the Federal University of Paraná (UFPR), Curitiba, Paraná, Brazil. Table 1 shows the list of voucher specimens deposited at the Herbarium of the Botany Department, UFPR, and the seeds used for the morphological and morphometric studies for all species, and for asymbiotic germination.

### Morphological and morphometric analysis of seeds

An average of 30 seeds per specimen was analyzed using a light (Fig 2C) and a scanning electron microscope (SEM). For SEM observations, the seeds were fixed with double-sided carbon tape and coated with gold. Qualitative data on the general seed morphology, including color, ornamentation, shape, micropillar opening, and the presence of a cuticular deposit, were analyzed. Qualitative and quantitative analyses were performed using a light microscope

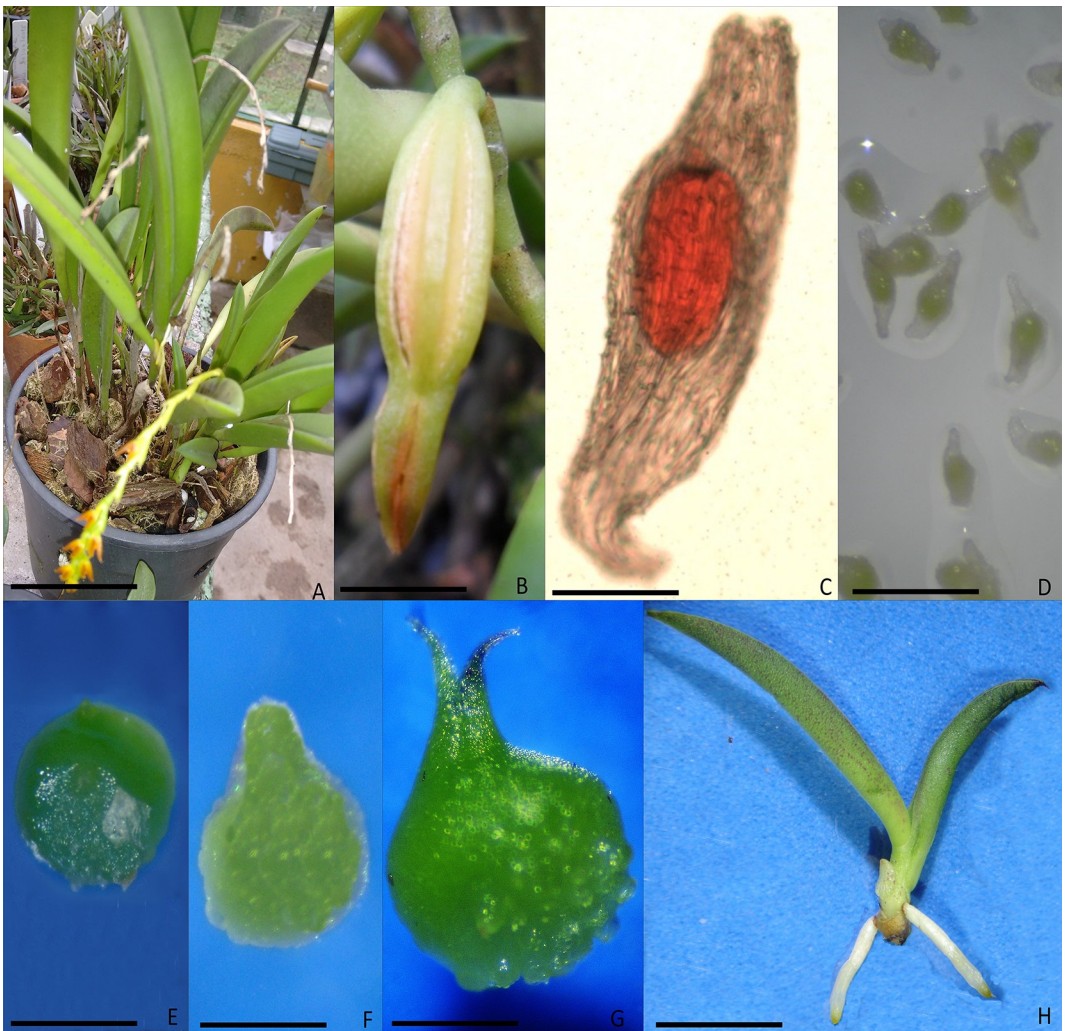

**Fig 2. Acianthera ochreata.** (A) Plant cultivated in the greenhouse, (B) dehiscent capsule, (C) seed observed in optical microscope, (D-H) *in vitro* germination, (D) seed with chlorophyllous embryo (4 days), (E) testa ruptured/ chlorophyllous protocorm (germination, 7 days), (F) protocorm with apex and / or rhizoids (14 days), (G) protocorm with two or more leaves (60 days), (H) seedling (120 days). Bar: A = 2 cm, B = 0,5 cm, C = 100 μm, D-G = 200 μm, H = 1cm.

(Olympus BX41 with DC30 camera), and the following variables were measured for seeds and embryos: length (L), width (W), L/W ratio, volume, and percentage of air space between the testa and the embryo. The width and length were measured with a micrometer at the longest and widest axis of the seed. Seed volume was calculated as $2 [(W/2)^2 \cdot (L/2) \cdot (\pi/3)]$. Embryo volume was calculated by using the formula $4/3\, \pi \cdot L/2 \cdot (W/2)^2$. The percentage of air space was calculated as [(seed volume—embryo volume) / seed volume] x 100. The terminology and methods adopted were those of: Arditti et al. [29, 30], Arditti and Ghani [24] and Barthlott et al. [25].

## Seed viability

**Tetrazolium test.**   Seeds (5 mg) were placed in a 1.5 mL Eppendorf microtube and pre-conditioned in a 10% sucrose solution at room temperature for 24 h. They were then immersed in a 1% tetrazolium solution for 24 h at 40˚C in a water bath in the dark. The

**Table 1. Species of the subtribe Pleurothallidinae used for morphological, morphometric, and *in vitro* germination analysis in this study.**

| Species | Voucher | *In vitro* germination |
|---|---|---|
| *Acianthera aphthosa* (Lindl.) Pridgeon & M. W. Chase | Koene, FM. 010 | Yes |
| *A. hatschbachii* (Schltr.) Chiron & van den Berg | Oliveira, LRL. 019 | No |
| *A. ochreata* (Lindl.) Pridgeon & M.W. Chase | Koene, FM. 005 | Yes |
| *A. prolifera* (Herb. ex Lindl.) Pridgeon & M. W. Chase | Koene, FM. 001 | Yes |
| *A. sonderiana* (Rchb.f.) Pridgeon & M.W. Chase | Koene FM. 009 | No |
| *Anathallis obovata* (Lindl.) Pridgeon & M.W. Chase | Santos, MC. 020 | Yes |
| *Dryadella liliputiana* (Coan.) Luer | Imig, DC. 381 | Yes |
| *D. zebrina* (Porsch) Luer | Imig, DC. 405 | No |
| *Octomeria gracilis* Lodd. ex Lindl. | Koene, FM. 014 | Yes |
| *Pabstiella fusca* (Lindl.) Chiron & Xim. Bols | Koene, FM. 013 | Yes |
| *Stelis grandiflora* Lindl. | Ignowski, H. 011 | Yes |

solution was drained from the tubes with a micropipette, and the seeds were washed twice with distilled water, following the methodology proposed by Hosomi et al. [31]. Red seeds were classified as viable and used for morphometric studies.

***In vitro* germination.** Seeds were surface-sterilized by dipping into a 1% sodium hypochlorite (10–12% PA) solution (NaClO) (v / v) containing 0.1% Tween 20® (v / v) for 15 min while stirring. The seeds were then transferred to a funnel coated with sterile filter paper and washed six times with sterile distilled water. The seeds were dried on sterile filter paper and inoculated in Petri dishes (150 mm x 20 mm) containing 30 mL of the woody plant medium (WPM) [14]. The media were supplemented with 5.6 g $L^{-1}$ agar from HiMedia® (Mumbai, India) and 3% sucrose (w/v). The pH of the media was adjusted to 5.8 with 0.1 N NaOH or HCl before the addition of agar. Culture media were sterilized by autoclaving for 20 min at 121˚C.

For the evaluation of seed germination about 500 seeds per Petri dish, with three Petri dishes per species, were inoculated. Five fields with 100 seeds per plate were marked and protocorm development was evaluated from two to 16 weeks based on the following stages: 1, seed with chlorophyllous embryo (Fig 2D); 2, testa ruptured/chlorophyllous protocorm (germination) (Fig 2E); 3, protocorm with apex and/or rhizoids (Fig 2F); 4, protocorm with one or two leaves (Fig 2G); 5, protocorm with two or more leaves and root (seedling) (Fig 2H). The germination rate was evaluated after four, eight, and 12 weeks of cultivation, and the average time (in days) to reach the stages was calculated for 24 weeks. The Petri dishes with seeds were maintained at 26±2˚C/18±2˚C (day/night), with a 16 h photoperiod under fluorescent lamps at a light intensity of 40 μmol $m^{-2}$ $s^{-1}$.

After seedling formation, the radicles can be cut, and explants cultured in a medium containing cytokinins to induce shoots (Fig 5C) or explants can be subcultured in flasks containing a medium supplemented with activated charcoal, where elongation and root development occur (Fig 5D). Finally, *A. ochreata* and *A. prolifera* plants were transplanted into sowing trays (3.5 $cm^2$) containing commercial Forth® substrate, composed of a mixture of coconut fiber, *Pinus* bark, and charcoal with fine vermiculite Eucatex® (1:1) (v/v). The seedlings were acclimatized in a greenhouse at room temperature (25 ± 2˚C day / 20 ± 2˚C night), under a photoperiod of 12 h and a light intensity of 50 μmol·$m^{-2}$·$s^{-1}$

## Statistical analysis

The experimental design was completely randomized. The data of the frequency (%) of the developmental protocorm stages were submitted to the Bartlett and the Shapiro-Wilk

**Table 2. Morphological seed characteristics of species belonging to the subtribe Pleurothallidinae.**

| Species | Color | Shape | Testa cell | Micropillar opening | Ornamentation |
|---------|-------|-------|-----------|---------------------|---------------|
| *Acianthera aphthosa* | Pale Yellow | Fusiform | Oblong | Yes | Papillae |
| *A. ochreata* | Brown | Ellipsoid | Oblong | Yes | Verrucosities |
| *A. hatschbachii* | Brown | Filiform | Oblong | Yes | Papillae |
| *A. prolifera* | Pale Yellow | Fusiform | Hexagonal | Yes | Absent |
| *A. sonderiana* | Brown | Clavate | Oblong | Yes | Verrucosities |
| *Anathallis obovata* | Pale yellow | Ellipsoid | Oblong | Yes | Absent |
| *Dryadella liliputiana* | Brown | Ellipsoid | Oblong | Yes | Verrucosities |
| *D. zebrina* | Brown | Ellipsoid | Oblong | No | Verrucosities |
| *Pabstiella fusca* | Brown | Clavate | Oblong | Yes | Papillae |
| *Octomeria gracilis* | Brown | Fusiform | Oblong | Yes | Papillae |
| *Stellis grandiflora* | Pale yellow | Clavate | Oblong | Yes | Absent |

normality test and analysis of variance (ANOVA). The means were compared by the Tukey test at a level of significance of 5%. The statistical program used was PAST 3.3 software.

## Results

### Morphological and morphometric analysis of seeds

The seeds of the orchids studied exhibited diversity in their shape, size, volume, and seed coat (ornamentation) (Tables 2 and 3), as well as in their embryo morphometry (Table 4).

Pale yellow seeds were observed in *A. prolifera*, *D. liliputianauana*, and *S. grandiflora*, while the seeds of the other species studied were brown (Table 2).

The seeds had several shapes, including fusiform, filiform, ellipsoid, and clavate (Fig 3A–3K). Generally, cells were shorter at either pole, while the medial cells of the seed coat were elongated. The chalazal pole of the seeds was closed, and the micropillar end was open (Fig 3F), except for *D. zebrina*, which was also closed (Table 2). The testa cells observed were transparent, longitudinally oriented, and oblong (Fig 3L), except in *A. prolifera*, in which they were hexagonal (Fig 3M). The ornamentation pattern of the testa cells of some species included papillae or verrucosities (Fig 3N), though these were absent from *A. prolifera*, *A. obovata*, *S. grandiflora* and *A. hatschbachii*

**Table 3. Morphometric data of seeds of species belonging to the subtribe Pleurothallidinae.**

| Species | Length | Width | Length/width Ratio | Testa cells* | Volume |
|---------|--------|-------|--------------------|--------------|--------|
| | (µm) | (µm) | | length- width | mm³ x 10⁻³ |
| *Acianthera aphthosa* | 513 ±24 | 115±21 | 1.075 | 9–12 | 1.401 |
| *A. hatschbachii* | 401 ±27 | 111±31 | 1.386 | 6–12 | 1.293 |
| *A. ochreata* | 473 ±24 | 139±18 | 1.768 | 10–12 | 2.392 |
| *A. prolifera* | 742±119 | 135±43 | 2.700 | 14–10 | 3.540 |
| *A. sonderiana* | 307±81 | 159± 18 | 1.120 | 3–6 | 2.032 |
| *Anathallis obovata* | 244±31 | 109±17 | 1.101 | 4–6 | 0.759 |
| *Dryadella liliputiana* | 265±42 | 146±20 | 1.125 | 2–5 | 1.479 |
| *D. zebrina* | 262±17 | 154±20 | 1.720 | 2–3 | 1.626 |
| *Octomeria gracilis* | 247±28 | 90±16 | 1.084 | 5–6 | 0.524 |
| *Pabstiella fusca* | 187±31 | 83±7 | 0.140 | 4–3 | 0.337 |
| *Stelis grandiflora* | 195±19 | 127±21 | 1.053 | 2–5 | 0.823 |

The values represent the means and standard deviations.

*Testa cells: measurements on the largest length and width of the testa of the seed.

**Table 4. Morphometric data of embryos of species belonging to the subtribe Pleurothallidinae.**

| Species | Length | Width | Volume | Air space | SV/EV* |
|---|---|---|---|---|---|
| | (μm) | (μm) | (mm³ x 10⁻³) | (%) | |
| *Acianthera aphthosa* | 186±21 | 73± 14 | 0.5147 | 74.42 | 2.7220 |
| *A. hatschbachii* | 97±15 | 70± 15 | 0.2487 | 80.77 | 5.1990 |
| *A. ochreata* | 175±23 | 99±14 | 0.8976 | 62.48 | 2.6729 |
| *A. prolifera* | 216±47 | 80±9 | 0.7235 | 79.56 | 4.8929 |
| *A. sonderiana* | 84± 7 | 75±18 | 0.2473 | 87.83 | 8.2167 |
| *Anathallis obovata* | 76±17 | 69±6 | 0.1894 | 75.04 | 4.0074 |
| *Dryadella liliputiana* | 90±12 | 80±9 | 0.3014 | 79.61 | 4.9071 |
| *D. zebrina* | 84±7 | 79±7 | 0.2744 | 83.13 | 5.9256 |
| *Octomeria gracilis* | 77±7 | 71±5 | 0.2031 | 61.21 | 2.5800 |
| *Pabstiella fusca* | 69±8 | 64±7 | 0.1479 | 56.14 | 2.2785 |
| *Stelis grandiflora* | 79±5 | 75±7 | 0.2326 | 71.75 | 3.5382 |

Values represent means and standard deviation.

* seed volume (SV) to embryo volume (EV) ratio

(Fig 3O) (Table 2). The number of testa cells ranged from 2 to 14 on the longest axis and 3 to 12 on the widest axis, with the highest number in *A. prolifera* and *A. ochreata* (Table 3).

The seeds showed high diversity in their size (length, width, and volume) (Table 3), as well as in the size and number of cells constituting the embryo (Table 4), despite their microscopic nature. The seed length ranged from 187±31 μm (*P. fusca*) to 742±119 μm (*A. prolifera*), while the width ranged from 83±7 μm (*P. fusca*) to 159± 18 μm for *A. sonderiana* (Table 3). *A. prolifera* and *A. ochreata* had length-to-width ratios of 2.700 and 1.767, respectively, while the L/W ratio in *P. fusca* was 0.140 (Table 3). *A. prolifera* had the largest seed volume (3.540 mm³ x 10⁻³) while the smallest was recorded in *P. fusca* (0.337 mm³ x 10⁻³) (Table 3).

The embryos were generally ellipsoidal and located at the center of the seed. Variation in length, width, and volume was also observed. The largest length was recorded for *A. prolifera* while *A. ochreata* had the largest width and volume, and *P. fusca* embryos had the smallest length, width, and volume (Table 4).

The largest percentage of air space was observed in *A. sonderiana* (87.83%), whereas the lowest percentage of air space was observed in *O. gracilis* (61.21%) and *P. fusca* (56.14%) (Table 4). The seed volume to embryo volume ratio was highest (8.2167) in *A. sonderiana*, followed by *D. zebrina* (5.9256), and *A. hatschbachii* (5.1990) and lowest in *P. fusca* (2.2785) (Table 4).

## Anatomical analysis

*Acianthera prolifera*, *A. obovata*, *O. gracilis*, and *P. fusca* had a thin testa (0.01 μm), while the testa was thick in other species (0.03 μm). The cells of the apical pole of the embryos of *A. aphthosa*, *A. ochreata*, *A. prolifera*, and *D. lilliputiana* were smaller than those of the basal pole, while these cells were similar in size in the remaining species. Most of the species studied did not have a suspensor, which occurred only in *A. prolifera* and *D. lilliputiana* (Table 5). The presence of a cuticle surrounding the embryo was observed in *A. aphthosa* and *P. fusca*, though none of the species had a cuticle around the testa.

## *In vitro* germination

The seeds of all of the micro-orchid species studied exhibited chlorophyllous embryos after three to eight days of sowing (Fig 4A). The first asymbiotic germination responses (seeds with

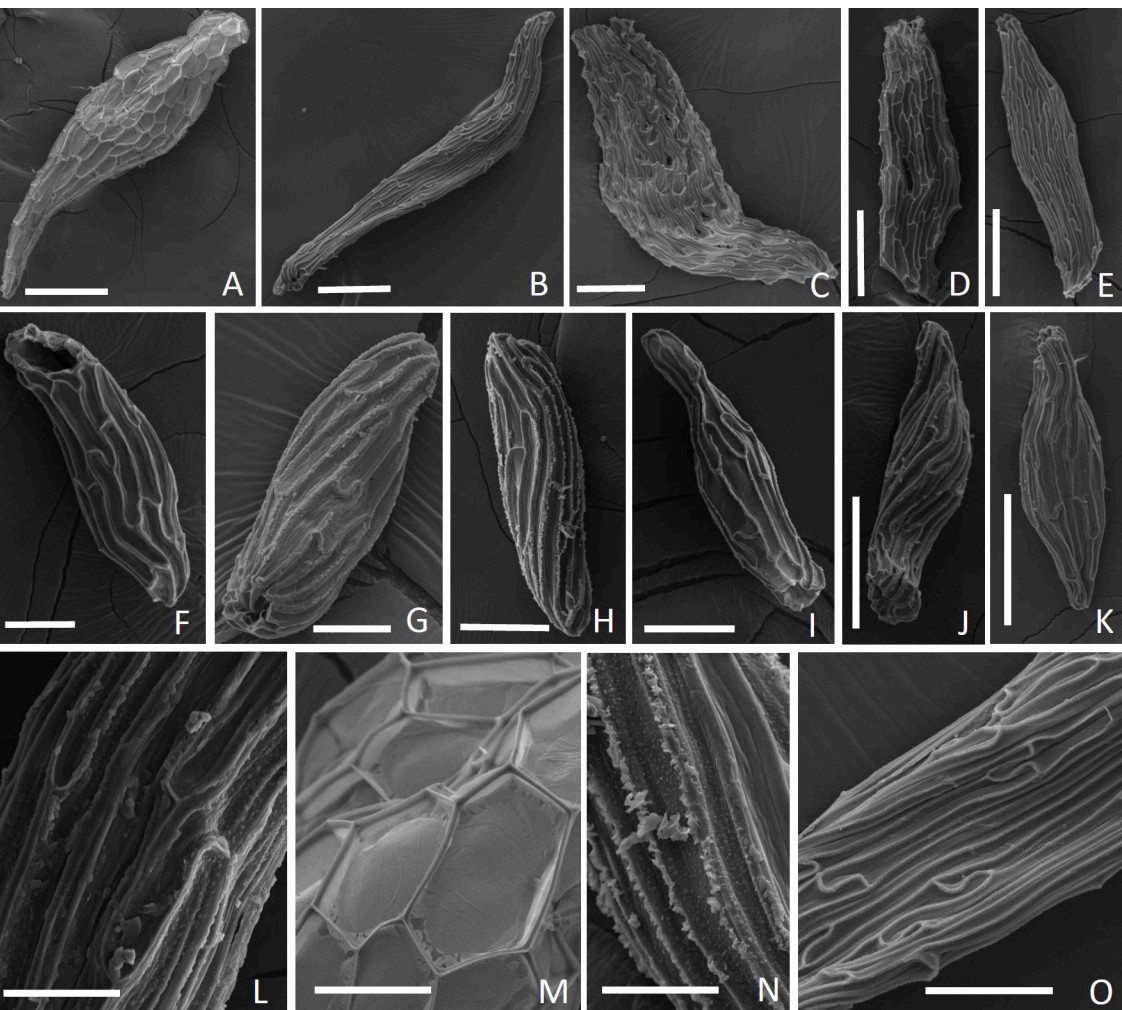

**Fig 3.** Seeds of Pleurothallidinae species observed in scanning electron microscopy: variation of seeds shape (A-K), (A) *Acianthera prolifera*, (B) *A. hatschbachii*, (C) *A. ochreata*, (D) *A. aphtosa*, (E) *A. sonderiana*, (F) *Anathallis obovata*, (G) *D. liliputiana*, (H) *Dryadella zebrina*, (I) *Pabstiella fusca*, (J) *Octomeria gracillis*, (K) *Stellis grandiflora*, (L) seed testa with oblong cells of *P. fusca*, (M) seed testa with hexagonal cells of *A. prolifera*, (N) testa cells of *D. zebrina* with ornamentation, (O) seed testa with smooth cells of *A. hatschbachii*. Bar: A-H = 50µm, I-O = 100µm.

ruptured testa and chlorophyllous protocorm) occurred after seven days of sowing for *A. prolifera* and *A. aphthosa*, and between nine and 13 days for the other species (Table 6). In general, most of the seeds had a low germination rate up to eight weeks after sowing, except for *A. prolifera* (76%). By 12 weeks after sowing, the highest germination rates were obtained for *A. prolifera*, followed by *A. ochreata* and *O. gracilis* (92%, 86%, and 77%, respectively). Of the species studied, only *A. prolifera*, *A. ochreata*, *A. aphthosa*, and *S. grandiflora* reached the seedling stage, with the time to this stage varying between four and 12 weeks (Table 6, Fig 5A and 5B).

The seeds of *A. obovata* and *D. liliputiana* showed similar behavior in the WPM, with germination rates lower than 40%, and reached the stage of protocorm with apex without the development of leaves and radicle, which became visible by 15–20 days, though they did not progress to the other stages by the 24-week evaluation. Likewise, *O. gracillis* protocorms did not develop leaves, although the germination rate was high (77%) at 12 weeks (Table 6). The seeds of *P.*

**Table 5. Anatomical characteristics of the seed testa of the embryo cells, and presence or absence of suspensor and cuticle in species belonging to the subtribe Pleurothallidinae.**

| Species | Testa | Embryo cells | Suspensor | Cuticle |
|---|---|---|---|---|
| | | Apical/basal pole | | |
| *Acianthera aphthosa* | Thick | Smaller/larger | Absence | Presence/embryo |
| *A. ochreata* | Thick | Smaller/larger | Absence | Absence |
| *A. prolifera* | Thin | Smaller/larger | Presence | Presence |
| *A. sonderiana* | Thick | Similar | Absence | Absence |
| *Anathallis obovata* | Thin | Similar | Absence | Absence |
| *Dryadella liliputiana* | Thick | Smaller/larger | Presence | Absence |
| *D. zebrina* | Thick | Smaller/larger | Absence | Absence |
| *Octomeria gracillis* | Thin | Similar | Absence | Absence |
| *Pabstiella fusca* | Thin | Similar | Absence | Presence/embryo |
| *Stelis grandiflora* | Thick | Similar | Absence | Absence |

*fusca* had the worst germination response, with only 4% germination by eight weeks after sowing. Furthermore, the protocorms did not develop in the WPM, and they were necrotic.

The analysis of the rates obtained at each stage of protocorm development confirmed that *A. prolifera* and *A. ochreata* had the best germination response (Fig 4). After two weeks, *A. prolifera* had the highest proportion of seeds with chlorophyll embryos (55%), followed by *A. ochreata* (52%) and *S. grandiflora* (49%) (Fig 4A). During this period, the highest germination rates (ruptured testa with chlorophyll embryo) occurred in *A. prolifera* (44%) and *A. ochreata* (39%) and were significantly higher than the other species analyzed, in which germination rates were below 10%.

By four weeks after sowing, all species except *P. fusca* had protocorms with apex (Fig 4B). By six to eight weeks after sowing, *A. prolifera* had the highest rate of protocorms with apex (27%) (Fig 4C); it was also the only species to have protocorms with leaves (21%) (Fig 4D). Seedling development began at 12 weeks, with the largest number of seedlings found in *A. prolifera* (32%), followed by *A. ochreata* (20%). The protocorms of *A. aphthosa* and *S. grandiflora* grew slowly, with less than 10% forming seedlings, though a longer period of evaluation (Fig 4E) or subculture to the same medium may be necessary. The development of the protocorms of *A. ochreata* into seedlings was somewhat slower than that of *A. prolifera*. After elongation, *A. ochreata* (Fig 5E) and *A. prolifera* (Fig 5F) were then successfully acclimatized in a greenhouse with 80% survival after three months or 12 months.

## Discussion

The species of the subtribe Pleurothallidinae showed considerable variation in the morphological characteristics of their seeds and embryos. The color of the seeds varied from pale yellow to brown. According to Swamy et al. [32], the microscopic size of orchid seeds makes it very difficult to visualize their color, but they are usually variations of yellow, brown, and white. Barthlott et al. [25] found that the most frequent color of seeds is brownish or dark brown, as observed in our study.

We observed seeds that were ellipsoid, clavate, fusiform, or filiform, shapes that have also been reported for other orchid species [30, 33–35]. Seed shape has an evolutionary significance, with fusiform seeds found in more primitive orchids and the various other forms in more evolved epidendroid orchids [36, 37]. Fusiform seeds in *A. aphthosa*, *A. prolifera*, and *O. gracilis* were also observed in this study.

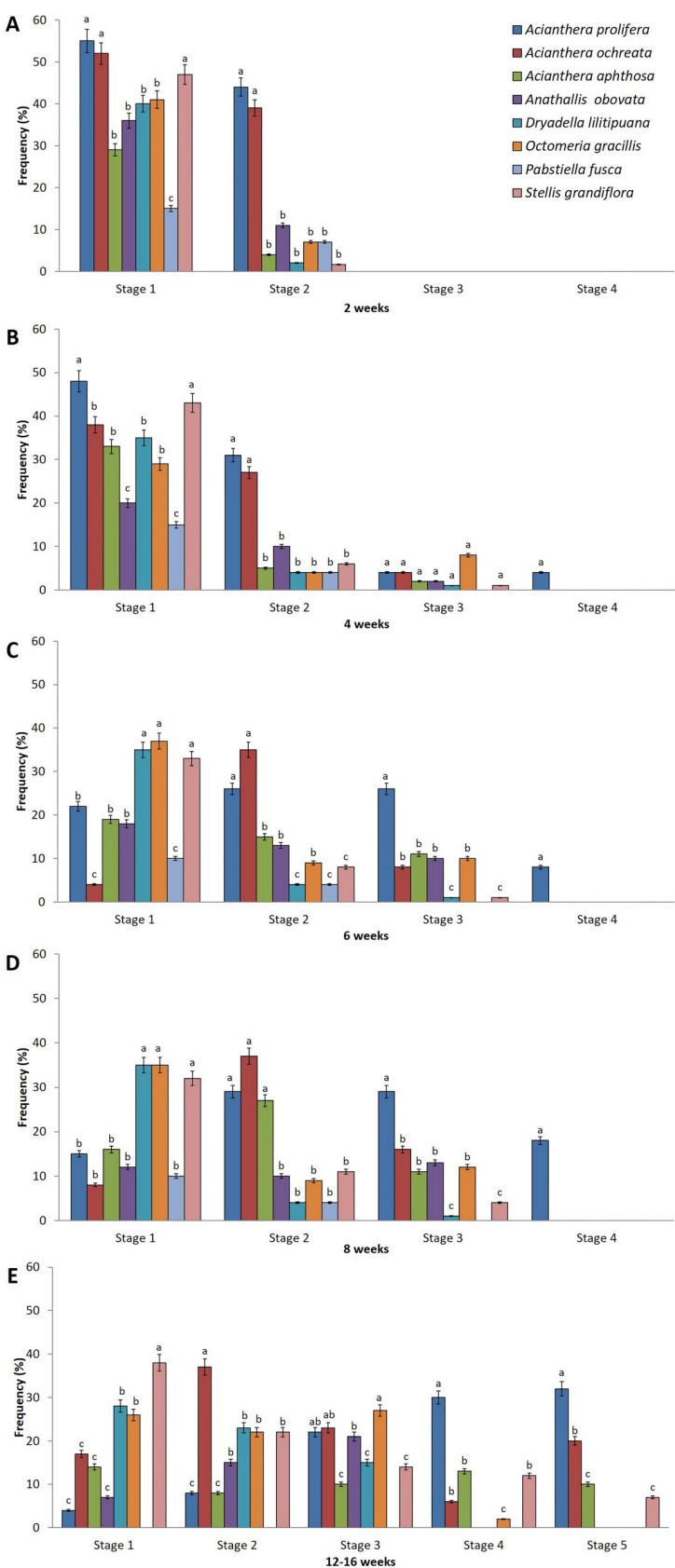

**Fig 4. Frequency (%) of the developmental protocorm stages of Pleurothallidinae species grown *in vitro* for two, four, six, eight, and 12 weeks in woody plant medium.** Stage 1, seed with chlorophyll embryo; 2, ruptured testa (germination); 3, protocorm with apex; 4, protocorm with one or two leaves; 5, protocorm with two or more leaves and root (seedling). Means followed by the same letter do not differ statistically by the Tukey test at 1% probability.

The testa in the middle part of the seed tended to be oblong or rectangular with elongated cells in the longitudinal axis of the seed [25], though *A. prolifera* cells were hexagonal and *A. hatschbachii* cells were linear. The testa cells of *A. obovata*, *A. prolifera*, and *S. grandiflora* did not show ornamentation, while papillae or verrucosities were apparent in the seeds of the other species. The presence or absence of ornamentation is cited in the literature to delimit some genera of Orchidaceae [33]. This is not the case in our study since among the species of *Acianthera*, *A. prolifera* had no ornamentation, *A. ochreata*, and *A. sonderiana* had verrucosities, and *A. aphthosa* and *A. hatschbachii* had papillae. Thus, the presence of seed ornamentation did not affect the asymbiotic germination of micro-orchids.

Variations in seed size and embryos occurred in the genera studied. In our study, *A. aphthosa* and *A. prolifera* had medium-sized seeds (500–900μm) according to the classification of Barthlott et al. [25]. Meanwhile, *A. hatschbachii*, *A. ochreata*, *A. sonderiana*, *A. obovata*, *D. zebrina*, and *O. gracilis* had small (200–500 μm) seeds, while *P. fusca* and *S. grandiflora* had very small seeds (100–200 μm) [25]. The seeds and embryos of *A. prolifera* and *A. ochreata* were larger (length, width, L/W ratio, and volume) and had higher rates of asymbiotic

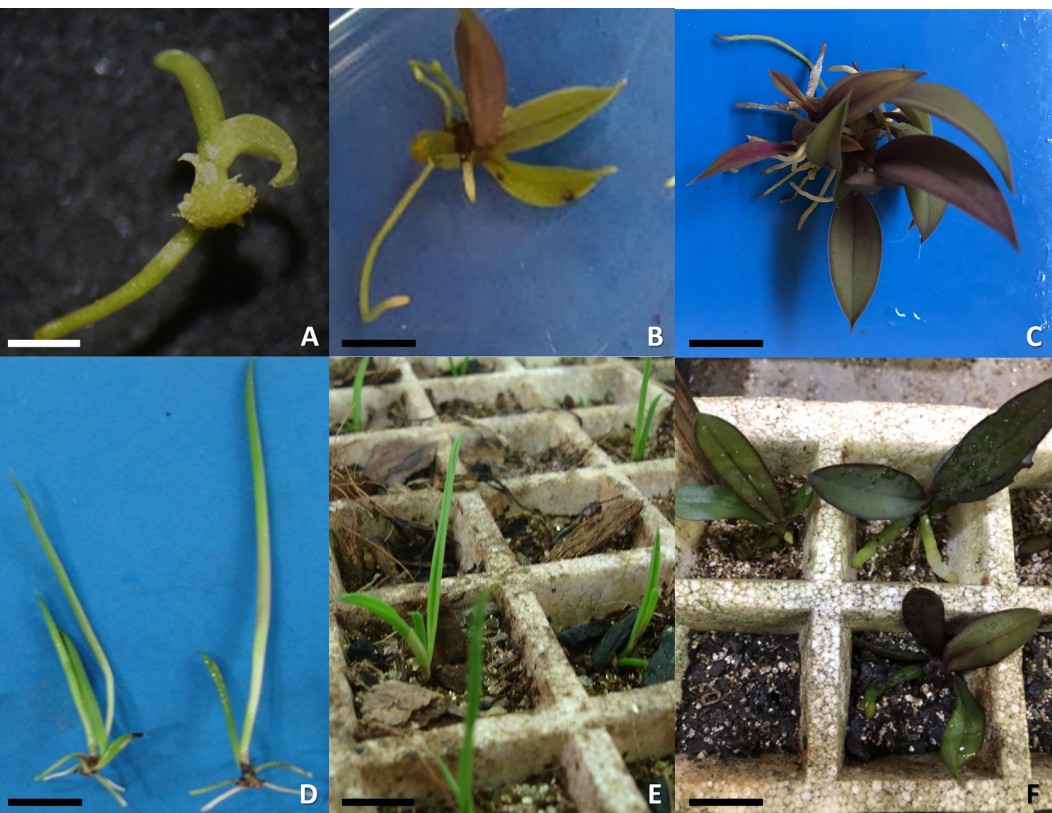

**Fig 5. *In vitro* germination of micro-orchids.** (A) *Stelis grandiflora* seedling, (B-C) *Acianthera prolifera* seedling, (D) *Acianthera ochreata* before transplant, (E) *A. ochreata* after acclimatization, (F) Acclimatizated plants of *A. prolifera*. Bar: A = 0,5 cm, B-D = 1,0 cm, E- F = 2 cm. (E) Photo by Damaris Lessmann.

**Table 6. Germination rate after four, eight, and 12 weeks in woody plant medium and the average time (days) for the protocorms development of the Pleurothallidinae species.**

| Species | Germination* (%) | | | Developmental stages** (days) | | | | |
|---|---|---|---|---|---|---|---|---|
| | **4** | **8** | **12** | **I** | **II** | **III** | **IV** | **V** |
| *Acianthera aphthosa* | 7 | 38 | 41 | 4 | 7 | 14 | 60 | 121 |
| *A. ochreata* | 35 | 53 | 86 | 4 | 9 | 14 | 60 | 119 |
| *A. prolifera* | 39 | 76 | 92 | 3 | 7 | 10 | 35 | 90 |
| *Anathallis obovata* | 12 | 23 | 36 | 3 | 10 | 15 | | |
| *Dryadella lilliputiana* | 4 | 15 | 38 | 7 | 13 | 21 | | |
| *Octomeria gracilis* | 4 | 21 | 77 | 5 | 11 | 21 | | |
| *Pabstiella fusca* | 4 | 4 | | 8 | 12 | | | |
| *Stelis grandiflora* | 7 | 15 | 51 | 8 | 11 | 21 | 95 | 127 |

* Evaluation after four, eight, and 12 weeks of sowing.

**Evaluation performed 16 weeks after sowing: I, seed with chlorophyllous embryo; II, ruptured testa (germination); III, protocorm with apex; IV, protocorm with one or two leaves; V, protocorm with leaves and root (seedling).

germination. According to Arditti et al. [29], L/W ratios provide data on the relative degree of truncation. *A. prolifera* and *A. ochreata* had L/W ratios of 2.700 and 1.767, respectively, while *P. fusca* had an L/W ratio of 0.140. Seeds with an L/W ratio of under 6.0 are referred to as truncated seeds, while those with an L/W ratio of over 6.0 are referred to as elongated seeds [30]. Based on this classification, all seeds in our study are truncated. According to Arditti et al. [29], seed volume is a better measure of seed size in orchids, which is in line with the findings of our study. They also considered that the seed volume and seed size are directly proportional to each other.

The number of testa cells along the longitudinal axis of the seed was low (2–6) in *D. lilliputiana* (2), *D. zebrina* (2), *S. grandiflora* (2), *A. sonderiana* (3), *A. obovata* (4), *P. fusca* (4), *O. gracilis* (5), and *A. hatschbachii* (6). On the other hand, there was a high number (9–14) of testa cells in *A. aphthosa* (9), *A. ochreata* (10), and *A. prolifera* (14). According to Barthlott et al. [25], the number of testa cells is coupled to cell division [38], a pattern that probably arises from slow or interrupted division in the integuments following fertilization. In other genera, where cell divisions continue, the seed coats are composed of numerous small cells. Seeds with only a few cells (five or fewer) along the longitudinal axis are especially common in Orchidaceae [25]. This feature is useful to characterize clades, usually at the subtribe level, with a high number of testa cells as the ancestral condition [25]. In addition, the species with a higher number of testa cells, *A. ochreata* and *A. prolifera*, showed a higher germination rate and plantlet formation in comparison with the other species analyzed.

As with the seeds, the embryos of *A. prolifera* and *A. ochreata* were also longer and wider, with greater volume. These species had higher germination rates after 12 weeks (92% and 86%, respectively). A similar result was reported by Tsutsumi et al. [39], who found that the larger embryos of *Liparis fujisanensis* F. Maek. ex Konta & S. Matsumoto were able to germinate more rapidly than smaller embryos. According to Yeung et al. [40], orchid embryos have fewer cells and are smaller than other flowering plant embryos. This can be a result of cells having a prolonged cell cycle time. The limited number of cells produced may also be due to the early cessation of mitotic activities.

Another characteristic that varied among the species of micro-orchids was the percentage of air space between the seed coat and the embryo. The highest percentage was observed in *A. sonderiana* (87.83%) and the lowest in *O. gracilis* (61.21%) and *P. fusca* (56.14%). Güler [41] also reported that the percentage of air space in certain species of *Anacampis*, *Neotinea*, and

*Orchis* ranged from 56 to 80%. Similar results were obtained by Swamy et al. [32], who found air space percentages ranging from 86.29% in *Cymbidium bicolor* Lindl. to 47.39% in *Coleogyne breviscapa* Lindl. Seeds with higher percentages of air space are lighter and can be more buoyant, aiding in wind dispersal across large geographic areas [28, 42]. Similarly, the seed volume to embryo volume ratio (SV/EV) was highest in *A. sonderiana* and lowest in *P. fusca*. The SV/EV ratio varied significantly among the species of the genus *Acianthera*, ranging from 8.2167 in *A. sonderiana* to 2.6729 in *A. ochreata*. This characteristic did not have any effect on the germination responses of the studied species of micro-orchids.

In our study, a suspensor was only visible in the seeds of *A. prolifera* and *D. liliputiana*. Orchidaceae is comprised of species that may present or not a suspensor [40]. Suspensor plays an important role during embryonic development, facilitating the movement of nutrients from maternal tissues to the embryo [5]. This structure may have facilitated the higher germination rates of *A. prolifera* seeds since the germination response of *D. lilliputiana* was lower. Another difference was that the seed testa of *D. lilliputiana* was thicker, and the seeds smaller than those of *A. prolifera*.

Structural polarity can occur in orchid embryos of some species, with larger cells at the apex and smaller ones at the base of the embryo, while these poles are uniform in size in other species [5]. In this study, the size of the poles was different in five species. Of these, *A. prolifera* (92%) and *A. ochreata* (86%) had high germination responses. Germination rates were lower in other three species exhibiting polarity, *A. aphthosa* (38%) and *D. liliputiana* (38%). According to Yeung [5], the cells at the apical pole of the embryo will form a meristematic zone, and the basal cells are designated to house the symbiont upon seed germination. A similar result was observed in *Phalaenopsis amabilis* (L.) Blume, which also exhibited these differences at the poles of the embryo and germinated more easily [43]. Another study found no difference between the poles of *Calypso bulbosa* (L.) Oakes., which is considered a species of difficult germination [44]. The authors found the same results for *P. fusca* and *S. grandiflora*, which had lower germination rates and developed no seedlings during the observation period. Embryos of species with no differences in size at the cell poles require additional growth and development of the embryo itself. In nature, these requirements can be provided by mycorrhizae [40].

A cuticle was present around the embryo in *A. aphthosa* and *P. fusca*. According to Aybeke [42], cuticle formation can ensure the retention of moisture by the cells of the embryo and increase its viability, which the author observed in the embryo of *Himantoglossum robertianum* (Loisel.) P. Delforge.

The results of asymbiotic germination indicated the highest germination rates in *A. prolifera* (92%), *A. ochreata* (86%), and *O. gracilis* (77%) at 12 weeks after sowing in WPM. The first two species had larger seeds and embryos with structural polarity, and both had some embryos developing to the seedlings stage in 12 to 16 weeks of cultivation. Furthermore, the lack of an endosperm may result in the seed having a lower water holding capacity, resulting in a more rapid change in water content [40]. Meanwhile, seedlings did not develop in *O. gracilis*, despite the high germination rate, with no germinated protocorms exhibiting leaf development. Other species, such as *D. liliputiana* and *A. obovata*, had germination rates of 38% and 36%, respectively, at 12 weeks after sowing, with no protocorms forming seedlings by 24 weeks after sowing. In addition to the morphological and morphometric characteristics that varied widely among the species studied, WPM may not have provided ideal conditions for all species, or as it was observed in a previous study for *A. prolifera*, the medium MS/2 was better for initial germination and WPM for the formation of seedlings (79% germination after 12 weeks) [13]. According to Yeung et al. [40], initial media for germination may be adequate; however, for continual development, a more complex medium may be required. Alternatively, a subculture in the WPM could be necessary to stimulate the development of protocorms in other stages.

The WPM has a low concentration of salts, less total nitrogen, and less ammonium than MS [45]. According to Suzuki et al. [46], the composition of the culture medium is essential for the success of the germination of orchid seeds, with results varying significantly from one species to another. Cultivating the seeds in a medium with an adequate nutritional composition helps to increase germination and, consequently, to produce large numbers of plants, which contributes to the conservation of endangered species [46].

The species with the worst germination response was *P. fusca* (4%), the protocorms of which were all necrotic by the fourth week after sowing in the WPM. This species had the smallest seeds and embryos, a cuticle covering the embryo and no structural polarity. For a majority of orchid seeds, even though the seed coat is thin, the addition of secondary walls, phenolic substances, and cuticular materials offer additional protection to the embryo within [40], and this may have hampered *in vitro* germination of *P. fusca*.

The culture medium used may not have been appropriate for this species, or disinfesting seeds with 1% sodium hypochlorite may have had a very strong effect since the seeds had a thin testa, i.e., this process may have caused damage to the embryos. Another factor is that some micro-orchids develop protocorms very slowly. For example, *Anathallis adenochila* (Loefgr.) F. Barros was found to take 12 months to develop seedlings (> 0.5 cm in height) [21]. In another species of the Pleurothallidinae subtribe, *Restrepia brachypus* Rchb.f., germination ranged from 7.96% in MS medium to 53.05% in Western (W) medium [22]. In addition, the protocorms of *R. brachypus* were subcultured for W medium and supplemented with banana pulp (60 g L$^{-1}$) for seedling development.

Another factor that may have influenced the low germination rates and slow protocorm development of some micro-orchids in our study may be the degree of seed maturity in asymbiotic germination. Yeung et al. [40] recommend that mature seeds of epiphytic orchids must be used for asymbiotic germination. The seeds of *A. prolifera* and *A. ochreata* could have been more mature than those of *P. fusca* when they germinated *in vitro*. In addition, an appropriate composition of the culture medium is assumed to be essential for the successful germination of immature orchids [47]. Factors such as degree of maturity of micro-orchid seeds need to be studied further for a better understanding of the germination of species that showed slow development and did not form seedlings in the WPM.

This study demonstrated considerable variation in the morphology, morphometry, and germination rates of the studied species. Germination and seedling formation of *A. prolifera* and *A. ochreata* occurred successfully in the WPM. Since germination was asynchronous, other formulations of the culture medium or supplements should be tested for their ability to accelerate the production of seedlings. Another approach would be to perform subcultures in the sowing stage so that there is no depletion of media components. Furthermore, evaluations should be carried out for more extended periods since some species grew very slowly, including *A. obovata*, *D. liliputiana*, and *P. fusca*.

*A. ochreata* and *A. prolifera* elongated and developed roots in WPM medium, supplemented with 1 g L$^{-1}$ activated charcoal (Fig 4D), as recommended by Koene et al. [13] in a previous study with *A. prolifera*. Finally, *A. ochreata* and *A. prolifera* achieved high survival rates [80%] after 12 months, when using commercial Forth® substrates, composed of a mixture of coconut fiber, *Pinus* bark, and charcoal with fine vermiculite Eucatex® (1:1). Koene et al. [13] had also successfully acclimatized *A. prolifera* (95% survival, after three months) using the same substrate and conditions of greenhouse. Therefore, our results offer potential possibilities for reintroduction programs in the future that can play a key role in reducing the threat of extinction for species of micro-orchids.

## Conclusions

In this study, we demonstrate efficient methods of rapid germination and seedling development of different species of Atlantic Rainforest micro-orchids. Additional studies are needed to accelerate the propagation process for some of the species that germinated slowly or minimally in this study. However, the technique used may help in the massive and rapid propagation of certain species for reintroduction into degraded habitats.

## Acknowledgments

The authors would like to thank CNPQ for the Master's scholarship granted to Franciele Marx Koene.

## Author Contributions

**Conceptualization:** Érika Amano, Eric de Camargo Smidt, Luciana Lopes Fortes Ribas.

**Data curation:** Franciele Marx Koene.

**Formal analysis:** Franciele Marx Koene, Érika Amano, Eric de Camargo Smidt, Luciana Lopes Fortes Ribas.

**Investigation:** Franciele Marx Koene, Érika Amano, Luciana Lopes Fortes Ribas.

**Methodology:** Franciele Marx Koene, Érika Amano, Eric de Camargo Smidt, Luciana Lopes Fortes Ribas.

**Supervision:** Érika Amano, Luciana Lopes Fortes Ribas.

**Validation:** Franciele Marx Koene, Érika Amano, Eric de Camargo Smidt, Luciana Lopes Fortes Ribas.

**Visualization:** Franciele Marx Koene, Érika Amano, Eric de Camargo Smidt, Luciana Lopes Fortes Ribas.

**Writing – original draft:** Franciele Marx Koene, Luciana Lopes Fortes Ribas.

**Writing – review & editing:** Érika Amano, Eric de Camargo Smidt, Luciana Lopes Fortes Ribas.

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
