## [Decision Letter · Decision Letter 0]

18 Sep 2020

PONE-D-20-27111

Asymbiotic germination and morphological studies of seeds of Atlantic Rainforest micro-orchids (Pleurothallidinae)

PLOS ONE

Dear Dr. Ribas,

Thank you for submitting your manuscript to PLOS ONE. After careful consideration, we feel that it has merit but does not fully meet PLOS ONE’s publication criteria as it currently stands. Therefore, we invite you to submit a revised version of the manuscript that addresses the points raised during the review process.

We look forward to receiving your revised manuscript.

Kind regards,

Jen-Tsung Chen, Ph.D.

Academic Editor

PLOS ONE

Journal Requirements:

Reviewers' comments:

Reviewer's Responses to Questions

**Comments to the Author**

1. Is the manuscript technically sound, and do the data support the conclusions?

Reviewer #1: Yes

Reviewer #2: Partly

Reviewer #3: Yes

Reviewer #4: Partly

2. Has the statistical analysis been performed appropriately and rigorously? 

Reviewer #1: Yes

Reviewer #2: Yes

Reviewer #3: Yes

Reviewer #4: No

3. Have the authors made all data underlying the findings in their manuscript fully available?

Reviewer #1: Yes

Reviewer #2: Yes

Reviewer #3: Yes

Reviewer #4: No

4. Is the manuscript presented in an intelligible fashion and written in standard English?

Reviewer #1: Yes

Reviewer #2: Yes

Reviewer #3: Yes

Reviewer #4: Yes

5. Review Comments to the Author

Reviewer #1: Comments: PONE-D-20-27111

In this manuscript, authors reported Asymbiotic germination and morphological studies of seeds of Atlantic Rainforest micro-orchids (Pleurothallidinae). After perusing the full manuscript, it was found that few similar papers have been published on this topic and it seems that this manuscript presents some applicable information to wide readers.

However, before a final decision, following issues should be addressed:

1. In Abstract: Add few starts lines about the background information of the research;

2. Add some most relevant keywords in the manuscript;

3. Revise the reference format as per journal author guidelines.

Reviewer #2: Dear Editor,

The paper describe 'Asymbiotic germination and morphological studies of seeds of Atlantic Rainforest micro-orchids (Pleurothallidinae)'. Orchid seeds are recalcitrant and the authors attempted to germinate it asymbiotic way.

The paper does not include novelty and including simple petri germination studies and some morphometric measurements.

The authors did not mention if the material terrestrial orchids or epiphytic orchids. They did not search whole literatures on orchids seed germination.

I am against to publication of this paper in Plos One.

Reviewer #3: PLOS ONE

Manuscript Number: PONE-D-20-27111

Title: A symbiotic germination and morphological studies of seeds of Atlantic Rainforest micro-orchids (Pleurothallidinae)

Article Type: Research Paper

Mona soliman

Review Report

The initial research articles in PLOS ONE describes the goals and the theories, methodologies and results of research and analysis of primary and unpublished studies. Original study papers can also include validation studies and disconfirmation of results that allow the exclusion and/or reproducibility of previously published findings by hypothetical assumptions. The manuscript entitled “A symbiotic germination and morphological studies of seeds of Atlantic Rainforest micro-orchids (Pleurothallidinae)” is interesting and mentioned to efficient methods of rapid propagation of different species of Atlantic Rainforest micro-orchids.

I recommend some minor changes before the paper is accepted in this periodical.

Abstract

Abstract is well written and informative

Introduction

The introduction is clearly and constructively written

Materials and Method

Line 123-132:

References are needed of all the above qualitative and quantitative methods

Line 141-149:

You must be subtitled to this paragraph to note the implications that the reader needs to understand……….thanks for your effort

Line 161-166:

In the results chapter, add the figures and tables to the correct place.

Results

These are nicely written

Discussion

The whole chapter deals with its purpose. The parameter studies are explained logically, but I think there should be a small reduction in discussions as far as possible

Comments for editor

Thank you for taking my manuscript review into consideration

I just need to check the paper again, as it is good, and only need minor corrections to be published and made available to readers.

Reviewer #4: All considerations were realized directly in the attached and revised version.

In addition:

About statystical analysis there are a mistake in comparisons of different species in Table 6. This comparison is similar to compare the natality of 'cats and lyons'. Also, comparisons between sizes of 'cats and lyons' is not biologically interesting. Also, a description about how many plants were used for measurements were not provided. In addition, the authors needs clarify if the time of evaluation was considered as treatment in the analysis. Tukey tests are not sound interesting in these cases, but only the standard deviation could be used for these analysis.

6. PLOS authors have the option to publish the peer review history of their article (what does this mean?). If published, this will include your full peer review and any attached files.

Reviewer #1: No

Reviewer #2: No

Reviewer #3: No

Reviewer #4: **Yes: **Jean Carlos Cardoso

---

## [Author Response · Author response to Decision Letter 0]

3 Nov 2020

Responses to Reviewer 1 

1. In Abstract: Add few starts lines about the background information of the research

Answer: Your request was accepted, see lines 25-28

2. Add some most relevant keywords in the manuscript

Answer: Relevant keywords have been added in blue

Keywords: Orchidaceae; micromorphology; morphometry, in vitro germination; woody plant medium; SEM.

3. Revise the reference format as per journal author guidelines.

Answer: The references format was reviewed and corrected as guidelines of Plos One

Responses to Reviewer 3

Materials and Method

1. Line 123-132:

References are needed of all the above qualitative and quantitative methods

Answer: The references were added, see lines 154-156

2. Line 141-149:

You must be subtitled to this paragraph to note the implications that the reader needs to understand……….thanks for your effort

Answer: Your request was accepted, see lines 159 and 166

3. Line 161-166:

In the results chapter, add the figures and tables to the correct place.

Answer: This has been corrected in the manuscript.

4. Discussion

The whole chapter deals with its purpose. The parameter studies are explained logically, but I think there should be a small reduction in discussions as far as possible

Answer: Your request was accepted.

5- About statystical analysis there are a mistake in comparisons of different species in Table 6. This comparison is similar to compare the natality of 'cats and lyons'. Also, comparisons between sizes of 'cats and lyons' is not biologically interesting. Also, a description about how many plants were used for measurements were not provided. In addition, the authors needs clarify if the time of evaluation was considered as treatment in the analysis. Tukey tests are not sound interesting in these cases, but only the standard deviation could be used for these analysis.

Answer: Statistical analysis was removed from the table 6.

Responses to Reviewer 4 (J. C. Cardoso)

Abstract

1. WPM medium is not a conventional medium for orchid germination. Authors have some reference support the use of this culture for this purpose? Because germination of orchids depends strong from the mineral composition.

References and the justification for using this medium were included in the manuscript.

Answer: WPM medium was previously tested for Acianthera prolifera and compared with other formulations (MS, MS/2 and KC) and showed better response, accelerating seedling development, so it was used to evaluate the germination of the other 10 micro-orchid species. It was also recommended for other species of orchids, such as: Brasiliorchis picta and Hadrolaelia grandis when compared to conventional medium such as: MS, MS/2, KC and VW. 

The references are below and the justification for using the WPM medium were placed on the 

 lines 64-71 and 83-85 in the manuscript

Koene FM, Amano E, Ribas LLF. Asymbiotic seed germination and in vitro seedling development of Acianthera prolifera (Orchidaceae). South African Journal of Botany. 2019; 121:83-91.

Santos SA, Smidt EC, Padial AA, Ribas LLF. Asymbiotic seed germination and in vitro propagation of Brasiliorchis picta. African Journal of Biotechnology. 2016; 15:134–144.

Vudala SM, Ribas LLF. Seed storage and asymbiotic germination of Hadrolaelia grandis (Orchidaceae). South African Journal of Botany. 2017; 108:1–7.

2- What authors considering seedlings?

Answer: In our study is the stage that corresponds to protocorms with two or more leaves and roots (See Line 182). Some authors use plantlet for this stage. Seedlings are plants that develop from the embryo of a seed.

 3- “The protocorms of Anathalis obovata, Dryadella liliputiana, and Octomeria gracillis developed slowly in the WPM, not reaching the seedling stage in 24 weeks of cultivation”. Is this a problem of culture media use or from the species?

Answer: These species were tested only in this medium. To be sure if it is a problem of the medium or species, studies are needed comparing various formulations of culture medium or a subculture in the WPM could be necessary. This was also discussed in the lines 460-467. 

 The two studies on seed germination of the subtribe Pleurothallidinae species, reported percentage of germination at most 50% testing various formulations of culture medium and slow development of protocorms, requiring a subculture for the formation of seedlings. 

4- “This morphological and morphometric study contributes to the understanding of asymbiotic germination of some micro-orchid species” Please explain how?

Answer: The seeds and embryos of the species that germinated faster and with high percentages (A. prolifera and A. ochreata) presented some characteristics, such as: larger size and seed volume, structural polarity, higher number of testa cells and this may have contributed to this response. In our study the species with the lowest germination response was P. fusca (4%), this species had the smallest seeds and embryos, a cuticle covering the embryo and no structural polarity. We also observed that the percentage of air space between the seed coat and the embryo did not have effect on the germination responses of the studied species of micro-orchids. Further studies with other genera and species of the Subtribe Pleurothallidinae, and other culture media are needed to assess the influence of these morphological and morphometric characteristics, however, we achieved high germination percentages after six months of cultivation in WPM medium.

Introduction

5- Please the references follow numerical order, not alphabetic order. Please correct.

Answer: This has been corrected.

6- ... their high germination rates, which are commonly over 70%, as opposed to under 5% in ex vitro conditions. This is highly dependent on the species. Terrestrial orchids in general have many difficulties for germination using asymbiotic germination, compared with symbiotic. 

Answer: It was clarified in the text that this statement is for epiphytic species at line 62. Discussion with species of terrestrial orchids was also excluded.

7- What the reason for choice eleven species and these is specifical? Occurrence in region? 

Collection?

Answer: These species were previously pollinated and were selected for having mature capsules at the time of our study. Our goal was to compare the largest number of micro-orchid species, but we depended on the formation of capsules.

Material and methods

6- These species are uncommon! I recommended that the authors add a figure with the eleven species, if possible, at the reproductive stage.

Answer: A figure with the studied species was added as requested.

5- How was realized the pollination? Self-pollination? Crossing handling? Naturally?

If naturally, there are no possible hybridation?

Answer: Each species had three to four plants in a greenhouse, in which manual cross-pollination was performed.

6- How was the identification of species? Comparisons between the vouchers at herbarium? 

Answer: The species were identified by comparisons between the vouchers at herbarium and with the help of the specialist in this Subtribe (Pleurothallidinae): Eric de Camargo Smidt (co-author) of this manuscript. 

7- Recently legislation of Brazilian flora requires the registration of wild species in SISGEN system. Authors have these numbers for wild species collected in Brazil?

Answer: This registration is not required to studies with no accessions to the genes content or associated knowledge. Also, these plants came from outside Natural Protected Areas; they grow in periurban areas of Curitiba or purchased from reputable companies that produce and sell the species.

Results

8- 80% of survival for 3 months. For what species? In what environmental conditions? Please describe.

Answer: 80% of survival after three months for A. prolifera and A. ochreata. The environmental conditions in greenhouse were complemented. See lines 187-195 and 509-516.

Conclusions

Answer: They were corrected.

Figures 3 and 4

Very low quality figures

Answer: The figures were reviewed and prepared according to the guidelines of Plos One

---

## [Decision Letter · Decision Letter 1]

19 Nov 2020

Asymbiotic germination and morphological studies of seeds of Atlantic Rainforest micro-orchids (Pleurothallidinae)

PONE-D-20-27111R1

Dear Dr. Ribas,

We’re pleased to inform you that your manuscript has been judged scientifically suitable for publication and will be formally accepted for publication once it meets all outstanding technical requirements.

Kind regards,

Jen-Tsung Chen, Ph.D.

Academic Editor

PLOS ONE

Additional Editor Comments (optional):

Reviewers' comments:

Reviewer's Responses to Questions

**Comments to the Author**

1. If the authors have adequately addressed your comments raised in a previous round of review and you feel that this manuscript is now acceptable for publication, you may indicate that here to bypass the “Comments to the Author” section, enter your conflict of interest statement in the “Confidential to Editor” section, and submit your "Accept" recommendation.

Reviewer #2: All comments have been addressed

2. Is the manuscript technically sound, and do the data support the conclusions?

Reviewer #2: Yes

3. Has the statistical analysis been performed appropriately and rigorously? 

Reviewer #2: Yes

4. Have the authors made all data underlying the findings in their manuscript fully available?

Reviewer #2: Yes

5. Is the manuscript presented in an intelligible fashion and written in standard English?

Reviewer #2: Yes

6. Review Comments to the Author

Reviewer #2: Dear Editor

The MS is now ready for publication. I belive that they made all necesarry changes and additions on MS.

7. PLOS authors have the option to publish the peer review history of their article (what does this mean?). If published, this will include your full peer review and any attached files.

Reviewer #2: No

---

## [Editor Report · Acceptance letter]

4 Dec 2020

PONE-D-20-27111R1 

Asymbiotic germination and morphological studies of seeds of Atlantic Rainforest micro-orchids (Pleurothallidinae) 

Dear Dr. Ribas:

I'm pleased to inform you that your manuscript has been deemed suitable for publication in PLOS ONE. Congratulations! Your manuscript is now with our production department. 

Kind regards, 

on behalf of

Dr. Jen-Tsung Chen 

Academic Editor

PLOS ONE